# MEASURING GRAPH SIMILARITY USING TRANSFER COST OF FORSTER DISTRIBUTIONS

## ABSTRACT

In recent years, optimal transport-based distance metrics have shown to be effective similarity and dissimilarity measures for tackling learning problems involving network data. Prominent examples range from graph classification and community detection to object matching. However, the high computational complexity of calculating optimal transport costs substantially confines their applications to large-scale networks. To address this challenge, in this paper, we introduce a probability distribution on the set of edges of a graph, referred to as the *Foster distribution* of the graph, by extending Foster's theorem from electrical to general networks. Then, we quantify graph similarity between the corresponding Foster distributions by utilizing ideas from optimal transport theory. We also extend the proposed graph similarity measure to incorporate node features by defining an additional similarity measure on the node attributes. The applicability of the proposed approach is corroborated on diverse graph-structured datasets, through which we particularly demonstrate the high efficiency of computing the proposed graph distance for sparse graphs.

## 1 INTRODUCTION

Complex systems with multiple agents interacting are prevalent in nature and human society in different scales Brockmann & Helbing (2013); Strogatz (2001); Gao et al. (2016). The undesirable behavior of such systems, in the form of disease, economic collapse, and social unrest, has generated significant interest in the comparative analysis of networks. For example, comparing brain networks obtained from fMRI data is crucial for understanding the difference between healthy and ill patients, and comparing the trade networks of two countries is of importance in international economics Tantardini et al. (2019).

In recent years, graph distance metrics rooted in optimal transport (OT) have gained prominence due to OT's ability to capture both local and global graph characteristics, which is unattainable by mere comparison of adjacency matrices or examination of node-centric features such as degree distribution, degree variance, and clustering coefficients. While these OT-based distance measures are effective in tasks such as graph classification, their applicability in real-world networks—characterized by large scale—is impeded by substantial computational cost.

In this paper, we propose a new similarity measure that permits the comparison of large-size sparse networks in a computationally efficient manner to the existing optimal transport distance measures. The proposed similarity measure, *Graph Foster Distance* (GFD), combines Foster's theorem with the idea of optimal transport. Specifically, given two graphs, we propose to first estimate the foster distribution defined on the set of edges and then calculate the optimal transport-type distance between the evaluated distributions. Furthermore, we extend the applicability of GFD to encompass node features by introducing an additional transport distance that operates on the node attributes.

The formulation of GFD results in a simple linear program, the size of which is directly proportional to the square of the total edge count within the network. This results in a significant reduction of the computational cost, particularly for large-scale complex networks exhibiting sparsity. The enhanced computational efficiency of GFD is empirically demonstrated through comparative analyses with existing methods that leverage optimal transport on graph classification tasks. Our comprehensive analysis shows that GFD not only offers improved computational efficiency but also improves classification accuracy.

## 2 BACKGROUND

In this section, we review the essential background of graph theory, Resistance distance, and optimal transport that will aid in the development of GFD.

### 2.1 PRELIMINARIES

We denote an undirected graph by $\mathcal{G} = (V, E)$, where $V = \{v_1, \ldots, v_N\}$ is a finite set of $N$ vertices and $E \subseteq \{(v_i, v_j) \mid v_i, v_j \in V\}$ denotes the set of edges corresponding to unordered pairs of the elements of $V$. The adjacency matrix of the graph is denoted by $A \in \mathbb{R}^{N \times N}$, where $A_{ij} = 1$ if $(v_i, v_j) \in E$, otherwise $A_{ij} = 0$. The Laplacian of $\mathcal{G}$ is defined as

$$L = D - A, \tag{1}$$

where $D \in \mathbb{R}^{N \times N}$ is a diagonal matrix such that $D_{ii}$ is the degree of node $v_i$, $i = 1, \ldots, N$.

### 2.2 RESISTANCE DISTANCE

Given a graph $\mathcal{G} = (V, E)$, the resistance distance $d_{r,ij}$ between vertices $v_i$ and $v_j$ is defined as

$$d_{r,ij} = L_{ii}^\dagger + L_{jj}^\dagger - L_{ij}^\dagger - L_{ji}^\dagger, \tag{2}$$

where $L^\dagger$ is the pseudoinverse of the graph Laplacian matrix $L$. The Resistance distance is commonly studied in electrical engineering where it defines the resistance between two points in an electrical network when each edge is replaced by a unit resistor Bapat (2010). For an undirected network, the resistance distance is closely related to the Average Commute Time Distance (ACTD) and differs only by a constant. Specifically, ACTD between vertices $v_i$ and $v_j$, $d_{ij}$, is defined as

$$d_{ij} = V_G(L_{ii}^\dagger + L_{jj}^\dagger - 2L_{ij}^\dagger), \tag{3}$$

with $V_G(= |E|)$ is the number of edges in the graph. The ACTD between two vertices $v_i$ and $v_j$ measures the average number of steps a random walker, starting at node $v_i$, will take before arriving at $v_j$ for the first time, and going back. It has been shown that ACTD is sensitive to global structure change and insensitive to local changes, thus able to distinguish between trivial structural changes and significant structural changes Saerens et al. (2004). These aforementioned properties of ACTD are shared by resistance distance which makes resistance distance suitable for capturing the graph structure information.

### 2.3 OPTIMAL TRANSPORT

An optimal transport problem is concerned with transferring one probability distribution to another with the minimum effort. Specifically, let $\alpha$ and $\beta$ be two discrete probability measures defined on a metric space $(\mathcal{X}, d)$, then $\alpha = \sum_{i=1}^{n} a_i \delta_{x_i}$ and $\beta = \sum_{i=1}^{m} b_i \delta_{y_i}$, for some $x_1, \ldots, x_n, y_1, \ldots, y_m \in \mathcal{X}$ and $a_1, \ldots, a_n, b_1, \ldots, b_m \in [0, 1]$ such that $\sum_{i=1}^{n} a_i = 1$ and $\sum_{i=1}^{m} b_i = 1$. Then, the optimal transport problem from $\alpha$ to $\beta$ can be formulated as an optimization problem, given by,

$$\mathcal{L}_C(\mathbf{a}, \mathbf{b}) = \min_{\Pi \in U(\mathbf{a}, \mathbf{b})} \langle C, P \rangle,$$

where $\mathbf{a} = (a_1, \ldots, a_n)'$, $\mathbf{b} = (b_1, \ldots, b_m)'$, $U(\mathbf{a}, \mathbf{b})$ is the set of admissible couplings, i.e.,

$$U(\mathbf{a}, \mathbf{b}) = \{P \in \mathbb{R}_+^{n \times m} : P\mathbf{1}_m = \mathbf{a} \text{ and } P\mathbf{1}_n = \mathbf{b}\},$$

and $C$ is a cost function defined on $\mathcal{X}$.

## 3 PRINCIPLES OF GRAPH FOSTER DISTANCE

In this section, we first introduce Foster's theorem which allows us to interpret a graph as a probability measure and then define GFD over the Foster distributions induced by the graphs.

**Theorem 1.** (Foster's theorem) Let $d_{r,ij}$ be the resistance distance between vertices $v_i$ and $v_j$ of a connected graph $\mathcal{G} = (V, E)$, then $d_{r,ij}$ satisfies :

$$\sum_{(v_i, v_j) \in E} d_{r,ij} = N - 1 \tag{4}$$

The equivalent form is:

$$\sum_{(v_i, v_j) \in E} \frac{d_{r,ij}}{N - 1} = 1 \tag{5}$$

Foster's theorem Foster (1949) shows that for all connected graphs, the summation of all resistance distances between vertices pairs in the edge set divided by $N - 1$ are always 1. It's natural to extend the resistance distance on the edge set to be a discrete probability measure.

## 3.1 Foster Distribution on Graph

By leveraging Foster's theorem, we define the Foster Distribution on Graph and show that the Foster Distribution is a probability mass function on the edge set.

**Definition 1.** For a simple connected graph $\mathcal{G} = (V, E)$, the Foster Distribution $f_{\mathcal{G}}$ for graph $\mathcal{G}$ is a discrete probability mass function on Edge set $E$, defined as

$$f_{\mathcal{G}}((v_i, v_j)) = \frac{d_{r,ij}}{N - 1}, \qquad (v_i, v_j) \in E, \tag{6}$$

where $d_{r,ij}$ is the resistance distance between vertices $v_i$ and $v_j$.

**Theorem 2.** Foster Distribution $f_{\mathcal{G}}$ is a probability mass function on Edge set $E$.

*Proof.* To prove Function $f_{\mathcal{G}}$ is a probability mass function, we only need to show that:

$$f_{\mathcal{G}}((v_i, v_j)) \geq 0, \quad \sum_{(v_i, v_j) \in E} f_{\mathcal{G}}((v_i, v_j)) = 1 \tag{7}$$

Note that $f_{\mathcal{G}}((v_i, v_j)) \geq 0$ since $d_{r,ij} \geq 0$. From Foster's theorem, we also have

$$\sum_{(v_i, v_j) \in E} f_{\mathcal{G}}((v_i, v_j)) = \sum_{(v_i, v_j) \in E} \frac{d_{r,ij}}{N - 1} = 1$$

$\square$

## 3.2 Graph Foster Distance

Having obtained the foster distribution representing a graph, now we define the distance on the foster distributions of two graphs $\mathcal{G}_1$ and $\mathcal{G}_2$.

**Definition 2.** Given two connected undirected graphs $\mathcal{G}_1 = (V_1, E_1)$ and $\mathcal{G}_2 = (V_2, E_2)$, let $f_1, f_2$ be their respective Foster distributions, then the Graph Foster Distance $\text{GFD}(f_1, f_2)$ between $f_1$ and $f_2$ is defined as

$$\text{GFD}(f_1, f_2) = \min_{\pi \in \Pi(f_1, f_2)} \langle C, \pi \rangle = \sum_{i,j} C_{ij} \pi_{ij}, \tag{8}$$

where $\Pi(f_1, f_2)$ denotes the set of admissible couplings, i.e.,

$$\Pi(f_1, f_2) = \{\pi \in \mathbb{R}_+^{|E_1| \times |E_2|} \ : \ \pi \mathbf{1}_{|E_2|} = f_1 \text{ and } \pi^T \mathbf{1}_{|E_1|} = f_2\},$$

and $C \in \mathbb{R}_+^{|E_1| \times |E_2|}$ such that $C_{ij} = (f_1(i) - f_2(j))^2$.

Note that GFD has a similar structure to the Wasserstein Distance for discrete measures and can be easily estimated using linear programming solvers. This makes it computationally efficient for large sparse networks as the size of the linear program depends on the number of edges in the networks.

**Theorem 3.** The graph Foster distance satisfies

(1) Positive semi-definiteness: $\mathrm{GFD}(f_1, f_2) \geq 0$ and $\mathrm{GFD}(f_1, f_2) = 0$ if $f_1 = f_2$,

(2) Symmetry: $\mathrm{GFD}(f_1, f_2) = \mathrm{GFD}(f_2, f_1)$

for any Foster distributions $f_1$ and $f_2$.

*Proof.* The positive semi-definiteness directly follows the definition of GFD in (8). To be more specific, for any Foster distributions $f_1$ and $f_2$, each entry $C_{ij}$ in the cost function $C$ is nonnegative, together with the nonnegativity of the joint probability $\pi_{ij}$, we obtain $\mathrm{GFD}(f_1, f_2) \geq 0$. It can be observed that $f_1 = f_2$ implies $C = 0$, yielding $\mathrm{GFD}(f_1, f_2) = 0$ if $f_1 = f_2$. The symmetry follows from the observation that $\mathrm{GFD}(f_1, f_2) = \min_{\pi \in \Pi(f_1, f_2)} \langle C, \pi \rangle = \min_{\pi \in \Pi(f_1, f_2)} \langle C^T, \pi^T \rangle = \min_{\rho \in \Pi(f_2, f_1)} \langle C^T, \rho \rangle = \mathrm{GFD}(f_2, f_1)$, where $C^T$ and $\pi^T$ denote the transpose of $C$ and $\pi$, respectively. □

This theorem implies that the Foster graph distance, although may not be a metric on the space of discrete probability distributions, can be used as a measure to quantify the similarity between two graphs.

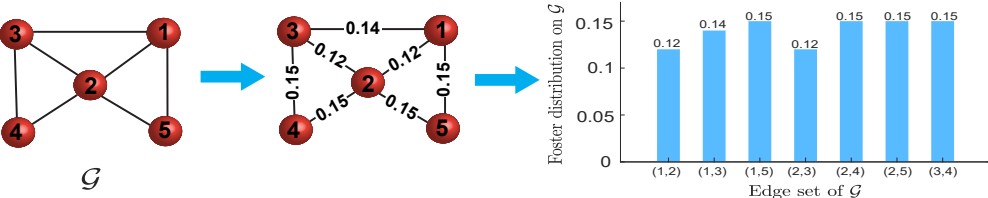

Figure 1: Give a graph $\mathcal{G}$, each edge is assigned a weight defined by the normalized resistance distance. These normalized resistance distances are then used to define Foster's distribution on the edge set.

### 3.3 GFD FOR GRAPHS WITH NODE ATTRIBUTES

The GFD defined in the previous subsection only takes into account the structure of the graphs. In several applications, graphs are usually found with node attributes, for example, brain networks, social networks, and chemical compounds Vayer et al. (2020). These node features can be readily incorporated into our GFD definition by adding an extra cost for the transportation of node features.

To begin, we consider undirected connected graphs with node features given by $\mathcal{G} = (V, E, A)$, where $A \in \mathbb{R}^{d \times N}$ contains the $d$-dimensional features of $N$ nodes. Now given two graphs $\mathcal{G}_1$ and $\mathcal{G}_2$ with their respective node features $A_1$ and $A_2$, we define the transport distance between node features as

$$\min_{\pi \in \Pi(D_1, D_2)} \langle C_a, \pi \rangle = \sum_{i,j} C_{a_{ij}} \pi_{ij},$$

where $\Pi(D_1, D_2)$ denotes the set of admissible couplings, i.e.,

$$\Pi(D_1, D_2) = \{\pi \in \mathbb{R}_+^{|V_1| \times |V_2|} \ : \quad \pi \mathbf{1}_{|V_2|} = D_1 \text{ and } \pi^T \mathbf{1}_{|V_1|} = D_2\},$$

with $D_1 = \frac{1}{|V_1|}\mathbf{1}_{|V_1|}$ and $D_2 = \frac{1}{|V_2|}\mathbf{1}_{|V_2|}$. This ensures identical weight for all the nodes. The cost matrix $C_a$ is defined as the Euclidean distance between the node features, i.e., $C_{a_{ij}} = \|A_{1_i} - A_{2_j}\|_2^2$, where $A_{1_i}$ denotes the $i^{th}$ vector of feature matrix $A_1$, for the continuous node features.

Having defined the transport distance between the node features, the distance between the graphs $G_1$ and $G_2$ is defined as

$$\mathrm{GFD} \text{ (with labels)} = \alpha \min_{\pi \in \Pi(f_1, f_2)} \langle C, \pi \rangle + (1 - \alpha) \min_{\pi \in \Pi(D_1, D_2)} \langle C_a, \pi \rangle,$$

where $\alpha \in (0, 1)$. In our numerical analysis, we take $\alpha = 0.5$.

## 4 RELATED WORKS

In this section, we review some of the works that leverage the power of optimal transport for graph comparison and are closely related to our work.

**Gromov-Wasserstein Distance (GW):** In Mémoli (2011), the authors propose Gromov-Wasserstein Distance which have become a powerful tool to measure the dissimilarity between structured objects like graphs. The principle idea of GW is to interpret the graphs as a metric measure space, for example, a graph $\mathcal{G} = (V, E)$ is interpreted as a metric measure space $(\mathcal{X}, d_1, u)$ where $\mathcal{X} = (v_1, ..., v_N)$, $u(v_i) = \frac{v_i}{\sum_{i=1}^{N} v_i}$, and $d_i$ is the distance defined on $\mathcal{X}$. Now the distance between two graphs or two metric measure spaces $(\mathcal{X}, d_1, u)$ and $(\mathcal{Y}, d_2, v)$ is defined as

$$GW_p(u, v) = \left( \inf_{\pi \in \Pi(u,v)} \sum_{\substack{x, x' \in \mathcal{X} \\ y, y' \in \mathcal{Y}}} \left( |d_1(x, x') - d_2(y, y')| \right)^p \pi(x, y)\pi(x', y') \right)^{1/p}, \quad (9)$$

where $\Pi(u, v)$ is the set of admissible couplings.

**Fused Gromov-Wasserstein Distance (FGW):** An extension of GW to include the node features was proposed in Vayer et al. (2020). This technique combines the Gromov-Wasserstein Distance with the Wasserstein Distance and improves the performance of GW on node-feature datasets; however, the computational complexity is still high.

**Filter Graph Optimal Transport (fFOT):** It is an optimal transport-based distance that considers the probability distribution of filtered graph signals for graph comparison purposes Maretic et al. (2022). The authors address the problem of graph alignment by computing permutations of the graph that minimize the filter distance. This technique can only compare graphs with no node attributed.

**Linear Fused Gromov-Wasserstein Distance (Linear FGW):** This technique was proposed to reduce the computational cost of estimating pairwise distance by leveraging kernel matrix. Specifically, the primary idea is to embed the graph with node features into a linear tangent space through a fixed reference graph and then estimate the Euclidean distance between the embeddings Nguyen & Tsuda (2023).

## 5 EXPERIMENT ANALYSIS

Here, we present the ability of our proposed graph similarity measure to capture the critical global properties of the graphs. Furthermore, we demonstrate its effectiveness on real-world datasets in terms of Graph classification.

For our initial analysis, we consider four different types of modular networks, as shown in Figure 2. In these networks, taken from Petric Maretic et al. (2019), each network has primarily two modules, and the primary distinction arises from the number of links connecting the modules. The networks $\mathcal{G}_1, \mathcal{G}_3$, and $\mathcal{G}_4$ has three links connecting the modules, while $\mathcal{G}_2$ has only one edge. In addition, $\mathcal{G}_3$ and $\mathcal{G}_4$ are isomorphic.

Table 1: Distance between modular networks

| GFD | $\mathcal{G}_1$ | $\mathcal{G}_2$ | $\mathcal{G}_3$ | $\mathcal{G}_4$ |
|-----|------|------|------|------|
| $\mathcal{G}_1$ | 0 | 0.08 | 0.01 | 0.01 |
| $\mathcal{G}_2$ | 0.08 | 0 | 0.05 | 0.05 |
| $\mathcal{G}_3$ | 0.01 | 0.05 | 0 | $10^{-30}$ |
| $\mathcal{G}_4$ | 0.01 | 0.05 | $10^{-30}$ | 0 |

We estimate the pair-wise distance between these networks, shown in Table 1. The calculated distances reveal two important observations: (i) The distance between the isomorphic networks $\mathcal{G}_3$ and $\mathcal{G}_4$ is $\approx 0$, which highlights that GFD is not affected by the node alignment; (ii) The degree of graph similarity observed between the weakly and strongly coupled modules is more than the two strongly coupled modules, i.e., GFD$(\mathcal{G}_1, \mathcal{G}_2) >$ GFD$(\mathcal{G}_1, \mathcal{G}_3)$. The second observation underscores the fundamental capability inherent in our similarity measure, namely, its adeptness in discerning crucial network connections. Each network under consideration has an identical number of connections; however, it is the inter-module connections that result in higher GFD in comparison to the intra-module connections.

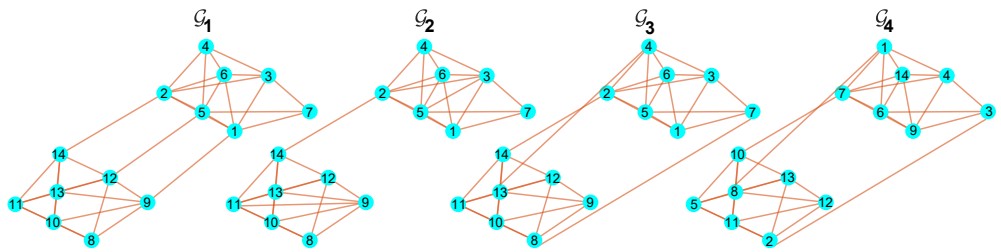

Figure 2: A set of four networks. The network $\mathcal{G}_1, \mathcal{G}_3$, and $\mathcal{G}_4$ has three links connecting the modules, while $\mathcal{G}_2$ has only one edge; $\mathcal{G}_3$, and $\mathcal{G}_4$ are isomorphic.

### 5.1 COMPARATIVE ANALYSIS

#### 5.1.1 DATASETS

We consider five widely used benchmark graph classification datasets: BZR Sutherland et al. (2003), MUTAG Debnath et al. (1991), PROTEINS Borgwardt & Kriegel (2005), AIDS Riesen et al. (2008), and REDDIT-BINARY(RB) Yanardag & Vishwanathan (2015). The details of these datasets including the number of graphs, average vertices, average edge number, number of classes, and node labels (Yes or No) are shown in Table 2. These datasets comprise both graphs with and without node attributes.

Table 2: Details of the experimental data sets.

| Dataset | # Graphs | # Avg. Nodes | # Avg. Edge | # Classes | Node Attr. |
|---|---|---|---|---|---|
| BZR | 405 | 35.75 | 38.36 | 2 | Yes |
| MUTAG | 188 | 17.93 | 19.79 | 2 | Yes |
| PROTEINS | 1113 | 39.06 | 77.82 | 2 | Yes |
| AIDS | 2000 | 15.69 | 16.20 | 2 | Yes |
| REDDIT BINARY(RB) | 2000 | 429.63 | 497.75 | 2 | No |

#### 5.1.2 EXPERIMENTAL DETAILS

We conduct a comparative analysis of our algorithm, evaluating both graph classification accuracy and the computational time. For this analysis, we consider four optimal transport-based methods: GW Mémoli (2011), fGOT Vayer et al. (2020), FGW Vayer et al. (2020), and Linear FGW Nguyen & Tsuda (2023), alongside with one non-OT-based method: FGSD Verma & Zhang (2017). Out of these five methods, only FGW and Linear FGW are equipped with the ability to incorporate node attributes and Linear FGW can only deal with continuous node attributes. The classification performance is evaluated using nonparametric 1-NN based on the estimated pairwise distance between the graphs for all the methods. For each dataset, we sample 100 graphs to generate the prediction accuracy in each experiment. Each experiment is repeated 30 times and the mean accuracy of the classification result is reported in Table 3. All the simulations are performed on a 3 GHz, 13th en Intel-Core i9-13900K with 48GB RAM machine.

For the graphs with discrete node attributes (MUTAG dataset), we first preprocess the discrete attributes using Weisfeiler-Lehman labeling (WL) by concatenating the labels of the node neighbors Vayer et al. (2020). This process is repeated $H$ times, which results in a $\mathbb{R}^{d \times H}$ feature matrix for each node in the network. The node feature cost matrix $C_a$ is then defined as the hamming distance between the node feature matrix Waggener & Waggener (1995). Note that Linear FGW can not be applied to MUTAG as the method is only designed for continuous node attributes. Similarly, we don't apply FGW, Linear FGW, and GFD (with labels) to the RB dataset since it does not have node features.

Table 3: Graph classification accuracy and computation time for the datasets in Table  2 where blue color indicates the best time computation results. 'OME' is out of memory error, '$> D$' denotes the computational time of more than a day, and 'NA' stands for not applicable. The bold fonts correspond to more than 3% improvement in classification accuracy.

|  | BZR | MUTAG | PROTEINS | AIDS | RB |
|---|---|---|---|---|---|
| FGSD | 83.62% (152.24s) | 80.00% (44.40s) | 54.85% (451.97s) | 95.25% (761.68s) | 50.00% (914.87s) |
| fGoT | 77.7% ($> D$) | 85.56% (1hr) | 64.56% ($> D$) | 99.0% ($> D$) | OME |
| GW | 79.60% (1665.84s) | 83.43% (74.64s) | OME | OME | OME |
| FGW | 82.47% 17.00s | 82.97% 2.30s | 65.53% 201.18s | 99.53% 243.76s | - |
| LinearFGW (with labels) | 66.22% 6.74s | NA | 51.67% 25.11s | % s | - |
| GFD (w/o labels) | 80.33% (3.25s) | 84.00% (0.79s) | 64.40% (7.59s) | 99.77% (17.77s) | **82.56%** (121.96s) |
| GFD (with labels) | **85.77%** 4.75s | 85.33% 2.91s | **70.03%** 17.05s | 99.60% 35.85s | - |

### 5.1.3 CLASSIFICATION RESULTS

Table 3 clearly indicates the **superior computational efficiency** of our algorithm for both cases: with and without incorporating node attributes. The run-time improvement is 2x-25x compared with other methods. We find that including node attributes improves our algorithm's classification accuracy without a significant increase in the computational time. In addition to the computational time improvements, our algorithm achieves **better classification accuracy** (more than 3% improvement). GFD performs even better on the Reddit-Binary dataset, achieving an improvement of **32.56%** over other methods. On the MUTAG dataset, we find fGOT to have slightly better accuracy, but it is 1200 times slower than GFD with labels. We also observe that Linear FGW improves the computational time of FGW, however, it results in degraded classification accuracy.

## 6 CONCLUSION

In this paper, we propose a foster distribution-based graph similarity measure, GFD, to address the high computational cost challenges associated with traditional methods using optimal transport. We utilize the idea of optimal transport to define the similarity measure between the graph foster distributions. The defined graph similarity measure results in a simple linear program with the number of constraints equal to the sum of the number of edges of two networks. This simple structure of the similarity measure results in not only lower computational cost than the existing methods but also improved graph classification accuracy as demonstrated by the extensive comparative analysis. We also extend GFD to include the node attributes while estimating graph similarity. The results in the comparative analysis show that adding node attributed improves the classification accuracy of GFD without a significant increase in the computational time.

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
