# OpenReview forum: "Measuring Graph Similarity Using Transfer Cost of Forster Distributions"
_ICLR.cc/2024/Conference — Submitted to ICLR 2024_

### Official Review · Reviewer_Gupv · 2023-10-23

**Soundness:** 1 poor
**Presentation:** 2 fair
**Contribution:** 1 poor
**Rating:** 3
**Confidence:** 4

**Summary:**

This paper proposes a new graph similarity measure based on the foster distribution and OT theory. The key idea is to represent graphs as a foster distribution over its edges, which is further used to define edge-wise distance and graph-level distance. A variant adopting node attributes is further proposed. Experiments on real world graph classification validate the effectiveness and efficiency of the proposed distance measure.

**Strengths:**

-	Conventional methods normally define graphs as distributions on nodes and seeks for alignment between nodes. The paper proposes a novel idea to represent graphs as distributions on edges, and tries to align edges instead.
-	A variant of GFD is proposed to handle graphs with node attributes.

**Weaknesses:**

-	Transport cost design: the cost design in Definition2 is sensitive to the scale of the graph, which is undesired. See details in the Questions section.
-	The connection between the proposed distance and the GW/FGW distance is unclear.
-	The GFD with labels solves node and edge coupling separately, which may lead to contradictory solutions (i.e., high node alignments with low edge alignments). FGW provides a better solution to optimize both coupling simultaneously.

**Questions:**

-	The proposed method is sensitive to the graph scale. I question the cost design $C_{ij}=(f_1(i)-f_2(j))^2$. In most OT applications, the pairwise cost is not dependent on the mass of two samples. However, the cost design in the paper is directly related to the mass on the samples. This may cause problem that $f_1(i)$ and $f_2(j)$ are in different scale (e.g., $G_1$ with N=10, but $G_2$ with N=1000).
-	What’s the adopted algorithm for solving the LP problem in Eq.(8)? What’s the corresponding complexity?
-	What’s the relationship between the proposed method and GW distance. Looking at Eq.(8), where GW is a quadratic term w.r.t. $\pi$ but the proposed method is linear w.r.t $\pi$.
-	For GFD with label defined at the end of page 4, the problem is solving two OT problems (one on edges and one on nodes). In my opinion, the proposed method is weaker than the FGW distance which optimizes the node distance (based on features) and connectivity (based on edges) simultaneously. For the proposed distance, there might be cases where two pairs of nodes (x,a) and (y,b) are well-aligned with high $\pi_1(x,a)$ and $\pi_2(y,b)$, but the edge alignment $\pi((x,a),(y,b))$ is low.
-	Experiments:
o	For the statement “GFD is not affected by the node alignment”, I think this is due to the OT theory, instead of the proposed GFD.
o	For table 1, it would be better to include additional analysis on distance between two graphs with quite different sizes. Since it seems your method is quite sensitive to the graph scale.
o	For section 5.1.2, why only 100 graphs are sampled for experiments? How are they sampled?
-	Writing: Section 3.3, Definition of graph as $G={V,E,A}$: A has already been defined as the adjacency matrix before, please change to another symbol.

---

### Official Review · Reviewer_XoAE · 2023-10-29

**Soundness:** 4 excellent
**Presentation:** 4 excellent
**Contribution:** 2 fair
**Rating:** 5
**Confidence:** 4

**Summary:**

This paper proposed a new graph similarity measure, Graph Foster Distance (GFD). The graph Foster distance is defined based on the optimal transport problem on the graph foster distributions. The Foster's theorem on the graph guarantees that the total resistance distances of all edges is n - 1. Thus, the resistance distances of edges normalized by n -1 provide a a distribution over edges, which is a graph Foster distribution. Then the graph Foster distance between two graphs is defined as the ultimate transport solution over two graph Foster distributions.

They first evaluate the graph Foster distance over synthetic modular graphs. They show that the graph Foster distance captures the global structure similarities between modular graphs. They also compare the graph Foster distance with other graph similarity measures on benchmark graph classification data sets. They show that the computation of graph poster distance is much faster compared with other methods. The graph Foster distance achieves better classification accuracy than other methods.

**Strengths:**

1. The graph is well-written and easy to read.

2. The paper proposed a new graph similarity measure based on the optimal transport problem. The experimental results show that the graph Foster distance is computationally efficient and achieves slightly better classification accuracy than other graph similarity measures.

**Weaknesses:**

1. The graph Foster distance is a direct application of the optimal transport problem on the graph Foster distributions.

2. Compared with the Fused Gromov-Wasserstein Distance (FGW), the improvement in the computation time and the classification accuracy for the graph Foster distance in the experiments is very marginal.

**Questions:**

1. In the experiments, the Fused Gromov-Wasserstein Distance (FGW) performs much faster than the Gromov-Wasserstein Distance (GW). FGW achieves very competitive performance compared with the graph Foster distance. Are there any potential explanations for this result?

---

### Official Review · Reviewer_zerX · 2023-10-31

**Soundness:** 2 fair
**Presentation:** 2 fair
**Contribution:** 2 fair
**Rating:** 3
**Confidence:** 4

**Summary:**

This article proposes a new measure of similarity between graphs that is based on optimal transport. The originality is that each graph is coded through the distribution of the resistance distance that is computed on each edge of the graph, and that the authors called the Foster distribution. Defining this Foster distribution of distances (on each edge) leads to a graph distance by using optimal transport to compare (and measure the similarity between) graphs using these distributions as marginals. Then it is used for some numerical experiments in graph classification.

**Strengths:**

- The concepts put forward in the article are simple, yet the parts about the resistance distance and the associated distribution are novel in this context.

- The work is reminiscent of the Wasserstein or the Gromov-Wasserstein distances applied to graphs, that were shown in the recent past to be efficient. Now the article brings the nice novelty of proposing a new coding of graphs through the Foster distribution. It is sound to try to see what happens with it.

- Some examples are in favour of using this novel distance.

**Weaknesses:**

- The numerical experiments are the weakest parts of the article. They only report on a graph classification task, on simple datasets were the accuracy properties of many methods are now comparable (so that there is a phenomenon of saturation on the learning tasks) ; and on a simplified situation with 4 graphs – Yet, one is not sure what should be the ground truth there.

- The authors should compare to a larger set of methods from the literature, for instance GED, several of the OT-based distances, other notions of distances between graphs (e.g. spectral notions) and on a larger set of graphs than the ones in Fig 2, so that the readers have more elements to apprehend how the new distance behave.

- It's the same for the experiments: the datasets are too simple, and of small size.

- The article does not refer properly to all the works using optimal transport and various ways one may code a graph, e.g. (the list is not complete; I only refer some works which propose relevant variations in the combination of OT and graphs):

. COPT, Dong & Sawin, 2020

. Optimal transport distances for directed, weighted graphs:... Nagai et al., arXiv:2309.07030

. Graph Diffusion Wasserstein Distances, Barbe et al., ECML-PKDD 2020

. Sobolev Transport, Le et al., AISTATS 2022

...

- There is no study of the possible equalities in Foster distributions that one could have without isomorphism of the graphs. Is it possible? Is there an y known result on that or prohibiting that ?


- The article is in truth not really dense, and maybe a more detailed study of this newly proposed distance would be better.
As it stands, it is one additional distance between graphs, and the readers don't have an actual reason to adopt the newly proposed one without more elements.

**Questions:**

- To compute the resistance distance, one needs to use the pseudo-inverse of the graph Laplacian matrix. Is it sufficiently stable ? The cost can be also computationally high. Does it limit the method to not so large graphs ?

- Are there theoretical results about cases of equality of the Foster distributions ?

- How does the method work on larger scale datasets ? (both larger graphs and datasets with more graphs) as compared to concurrent works ?

---

### Official Review · Reviewer_hTPf · 2023-11-02

**Soundness:** 2 fair
**Presentation:** 2 fair
**Contribution:** 2 fair
**Rating:** 3
**Confidence:** 3

**Summary:**

In this manuscript the authors proposed a new graph distance metrics based on resistance distance and Foster's theorem from electric engineering. Authors noted that this new graph distance measure has lower computational cost.

**Strengths:**

The Foster's theorem offers a way to normalize the resistance distances on a graph, and the authors further converted it to a probability mass function and constructed a new OT-based graph distance.

**Weaknesses:**

The paper seems half baked and lacks depth. For instance, Table 3 remains incomplete. The experiments are limited to (1) merely four toy graph examples with some quantitative comparisons and (2) a quick comparison on five graph classification beachmarks, as indicated in Table 3. The authors have failed to sufficiently explain or delve into the insights gained from their new graph distance measure. There's an absence of exploration regarding how it avoids the alignment problem, instead of merely asserting that "GFD is not affected by the node alignment" (page 5) based on four graph toy examples.

**Questions:**

1. Could the authors provide a more compelling argument in support of the proposed graph distance? Specifically, an elucidation on how it circumvents graph alignment would be great.
2. Table 3 appears to be incomplete. Given the authors' reference to reporting the mean after repeated runs, it would be beneficial to include the standard deviation. Additionally, there seem to be discrepancies in the figures for baseline models, such as fGOT, which show lower numbers than those reported in the original publication.

---

### Meta-Review · Area_Chair_nETn · 2023-12-08

**Metareview:**

The authors propose a new distance for graphs based on optimal transport between Forster distributions (i.e., a distribution on set of edges representation for each graph). The proposed distance is novel and is interesting. However, the Reviewers raised several concerns about the submissions, e.g., related works on optimal transport with graph structure, more rigorous experiments to illustrate advantages of the proposed distance as well as baselines. There is no rebuttal from the authors which makes those raised concerns remain questionable. Overall, I think the submission is not ready for publication yet. The authors can take into account of the Reviewers' comments to improve the work.

**Justification For Why Not Higher Score:**

+ The Reviewers raised several concerns about the proposed approach, especially about experiments: more rigorous experiment, and more baselines are required.

+ The Reviewers also raise the issue of related work on optimal transport with graph structure. It is better to place the contribution within the picture of OT and graph in the literature.

**Justification For Why Not Lower Score:**

N/A

---

### Decision · Program_Chairs · 2024-01-16

Reject